

# Complex versus simple models: ion-channel cardiac toxicity prediction

Hitesh B. Mistry

Division of Pharmacy, University of Manchester, Manchester, United Kingdom

## ABSTRACT

There is growing interest in applying detailed mathematical models of the heart for ion-channel related cardiac toxicity prediction. However, a debate as to whether such complex models are required exists. Here an assessment in the predictive performance between two established large-scale biophysical cardiac models and a simple linear model $B_{net}$ was conducted. Three ion-channel data-sets were extracted from literature. Each compound was designated a cardiac risk category using two different classification schemes based on information within CredibleMeds. The predictive performance of each model within each data-set for each classification scheme was assessed via a leave-one-out cross validation. Overall the $B_{net}$ model performed equally as well as the leading cardiac models in two of the data-sets and outperformed both cardiac models on the latest. These results highlight the importance of benchmarking complex versus simple models but also encourage the development of simple models.

## INTRODUCTION

There is a growing belief within the pharmaceutical industry that in order to improve predictions of future experiments more detailed mathematical models of biology are required (*Peterson & Riggs, 2015*; *Knight-Schrijver et al., 2016*). However by including more detail not only does the number of parameters that need to be estimated increase but so does the degree of structural uncertainty if the biology is not well understood i.e., the degree of confidence in the actual structure of the equations (*Beven, 2005*). The objective of this study is to look at this issue within the field of drug induced ion-channel cardiac toxicity. This area has a well-defined question relating to prediction where a debate about the complexity of the model needed is ongoing.

Numerous drugs were withdrawn from the market during the 1990s and early 2000s for causing a fatal arrhythmia, termed Torsades de Pointes (TdeP) (*Yap & Camm, 2003*). Current pharmaceutical industry screening strategies on identifying these compounds at an early stage in drug development are based on the following biological insights (*Antzelevitch & Sicouri, 1994*; *Witchel, 2011*). Prior to observing drug induced TdeP, prolongation of the QT interval is commonly seen within a patient. This prolongation is due to delayed repolarisation of cardiac cells within the ventricular wall, which is due to the drugs effect on the hERG ion-channel. Thus, the current approach in drug development involves

Corresponding author
Hitesh B. Mistry,
hitesh.mistry@manchester.ac.uk

screening a compounds effect against hERG in a high-throughput manner. However, there are other ion-channels involved in this process which the safety pharmacology community are now also screening compounds against (*Colatsky et al., 2016*). The question of interest then to the safety pharmacology community is: does measuring more than hERG improve prediction for TdeP, in humans?

In order to answer this question a clear definition of whether a compound has TdeP liabilities or not is required (*Wiśniowska & Polak, 2017*). The first study to examine the association between multiple ion-channel inhibition and TdeP risk (*Mirams et al., 2011*) used a database created by AstraZeneca (*Redfern et al., 2003*). This database was built using literature data only and has never been updated since its initial publication. More recent studies (*Kramer et al., 2013*; *Lancaster & Sobie, 2016*) have used the CredibleMeds database (*Woosley, Heise & Romero, 2017*; *Woosley et al., 2017*) which was formerly known as AzCERT. Their classification is based on an extensive search of both the literature and public databases and is continuously updated in-light of new evidence. Furthermore it is recognised by the clinical community unlike the AstraZeneca database.

In terms of the modelling approach used the literature is divided in terms of the complexity required (*Mistry, 2017*). The complex models used are based on biophysical models which describe the changes in ionic currents over time within a single cardiac cell (*Trayanova, 2011*). They contain 100s of parameters and 10s of differential equations. The drug input into these models involves scaling ion-channel conductance's by the amount of block at a given drug concentration (*Brennan, Fink & Rodriguez, 2009*). Two biophysical models that have gained favour in the literature are the *gold-standard*, as described by *Zhou et al. (2015)*, model by *O'Hara et al. (2011)*, herein referred to as ORD, which is being put forward for use by regulatory agencies (*Colatsky et al., 2016*) and another, by *Ten Tusscher & Panfilov (2006)*, forms a part of the *cardiac safety simulator* (*Glinka & Polak, 2015*), herein referred to as TT. An alternative simpler mechanistic model being put forward analyses the net difference, *via* a linear combination, in drug block of the ion-channels of interest, termed $B_{net}$ (*Mistry, 2017*). In that study $B_{net}$ gave similar performance to a joint three biophysical model/machine learning approach which used more than 300 metrics derived from the biophysical models (*Lancaster & Sobie, 2016*).

In this study the predictive performance of ORD, TT and $B_{net}$ models using a consistent and reliable definition of TdeP risk from CredibleMeds across three literature data-sets (*Mirams et al., 2011*; *Kramer et al., 2013*; *Crumb Jr et al., 2016*) was analysed. Two of these data-sets, *Mirams et al. (2011)* and *Kramer et al. (2013)*, measured drug effect against three ion-channels, hERG, Cav 1.2 and Nav 1.5 peak. The third and latest data-set, from *Crumb Jr et al. (2016)*, considers drug effect on 7 ion-channels, hERG (IKr), KCNQ1 + KCNE1 (IKs), Kv4.3 (Ito), Kir2.1 (IK1), Cav 1.2 (ICaL), Nav1.5 peak (INa) and Nav1.5 late (INaL), the largest number studied so far.

By using a consistent definition of TdeP risk across different data-sets that have different dimensionality in terms of ion-channels studied the analysis conducted will provide a detailed view on the performance of each model. Thus enabling scientists to make a more informed decision about which modelling approach is likely to be the most useful for the prediction problem considered.

**Table 1  Description of the two classification schemes constructed from the CredibleMeds database.**

| CredibleMeds | Description | QT/TdeP | TdeP |
|---|---|---|---|
| Known Risk (KR) | Known TdeP Risk | +ive | +ive |
| Possible Risk (PR) | Known QT Risk | +ive | −ive |
| Conditional Risk (CR) | Conditional TdeP Risk (e.g., drug-drug interaction) | −ive | −ive |
| No Risk (NR) | Not listed on CredibleMeds | −ive | −ive |

## METHODS

### Data

Ion-channel IC50 values, defined as concentration of drug that reduces the flow of current by 50%, were collected from three publications (*Mirams et al., 2011*; *Kramer et al., 2013*; *Crumb Jr et al., 2016*). Compounds within those data-sets were placed into two classification schemes based on the information in Credible Meds (*Woosley, Heise & Romero, 2017*; *Woosley et al., 2017*), see Table 1. The first classification scheme termed QT/TdeP focusses on both QT prolongation and TdeP risk, which was used in two previous studies (*Kramer et al., 2013*; *Lancaster & Sobie, 2016*). The second classification scheme focusses on known TdeP risk only. All data is provided in the Supplemental Information.

### Model input data

The percentage block against a given ion-channel inputted into all models was calculated using the mean maximal concentration observed corrected for plasma protein binding and is referred to as the effective therapeutic concentration (EFTPC), which was provided in the original articles, using a pore block model,

$$\%\mathrm{Block} = \frac{1}{1 + \frac{IC50}{EFTPC}}$$

## MODELS

### Single cell cardiac model simulations

The AP predict platform (*Williams & Mirams, 2015*) which is a web-based cardiac modelling simulation platform (https://appredict.cs.ox.ac.uk) was used to simulate the ORD and TT models in all cases except for one simulation study. A MATLAB version of the ORD model available on the Rudylab website (http://rudylab.wustl.edu) was used when simulating the block of seven ion-channels since that model on AP predict does not allow blocking of INaL—a current measured in the Crumb et al. data-set. The default settings within the AP predict platform were used i.e., 1 Hz pacing for 5 min with the APD90, time taken for the action potential to repolarise by 90%, recorded using the last cycle. The same protocol was applied in MATLAB when exploring the seven ion-channels within the ORD model i.e., 1 Hz pacing for 5 min with APD90 recorded using the last cycle. In all simulations drug block was initiated at the beginning of simulations.

### $B_{net}$

$B_{net}$ was defined as the net difference in block between repolarisation and depolarisation ion-channels as,

$$B_{net} = \sum_{i=1}^{n} R_i - \sum_{j=1}^{m} D_j$$

where $R_i$ and $D_j$ represent the percentage block against repolarisation and depolarisation ion-channels respectively for a specific drug. Ionic currents responsible for repolarisation are IKr IKs and Ito, and that for depolarisation are ICaL, INa (peak), INa (late) and IK1.

### Classification evaluation

For each compound the percentage change in APD90 compared to control (no block) from the biophysical model simulations was recorded as was the $B_{net}$ value. These values were then placed within a logistic regression analysis to assess their correlative value to either QT/TdeP or TdeP risk. This was done via a leave one out cross validation (LOOCV). This involves training a classifier to $n-1$ compounds and testing on the $n$th. Thus, all compounds perform part of the test-set. The predicted probability of risk for each test compound is then used to generate a ROC AUC (area under the receiver operating characteristic curve) which is reported. Note that LOOCV has been the method of choice within this field when assessing the correlation between metrics and drug risk (*Mirams et al., 2011*; *Kramer et al., 2013*; *Lancaster & Sobie, 2016*).

## RESULTS

### Data

The total number of compounds and their classification according to CredibleMeds across the three data-sets of interest can be seen in Fig. 1. Although the total number of compounds differs from one data-set to another the proportions that are KR, PR and CR/NR does not appear to.

The distribution of block against each ionic current, at the EFTPC, across all data-sets can be seen in Fig. 2. The plots show that the activity of the compounds is greatest against IKr across all data-sets. After IKr, ICaL appears to be the next channel for which a substantial amount of activity is seen. A somewhat surprising result is the degree of activity against INaL but not INa in the Crumb et al. data-set. The amount of activity against INaL in that data-set mirrors that of ICaL activity.

### Classification evaluation

The results of the leave-one-out cross validation for each data-set using various models for the two classification schemes can be seen in Tables 2 and 3. For the Mirams et al. data-set it's noticeable that ORD performs no better than using just block against hERG for either classification scheme. Furthermore for the QT/TdeP classification ORD is no better than random chance. Both TT and $B_{net}$ show a similar improvement over using just hERG block for both classification schemes.

Moving onto the Kramer et al. data-set the performance of all models improves dramatically over the Mirams et al. data-set. Here all three models show superior
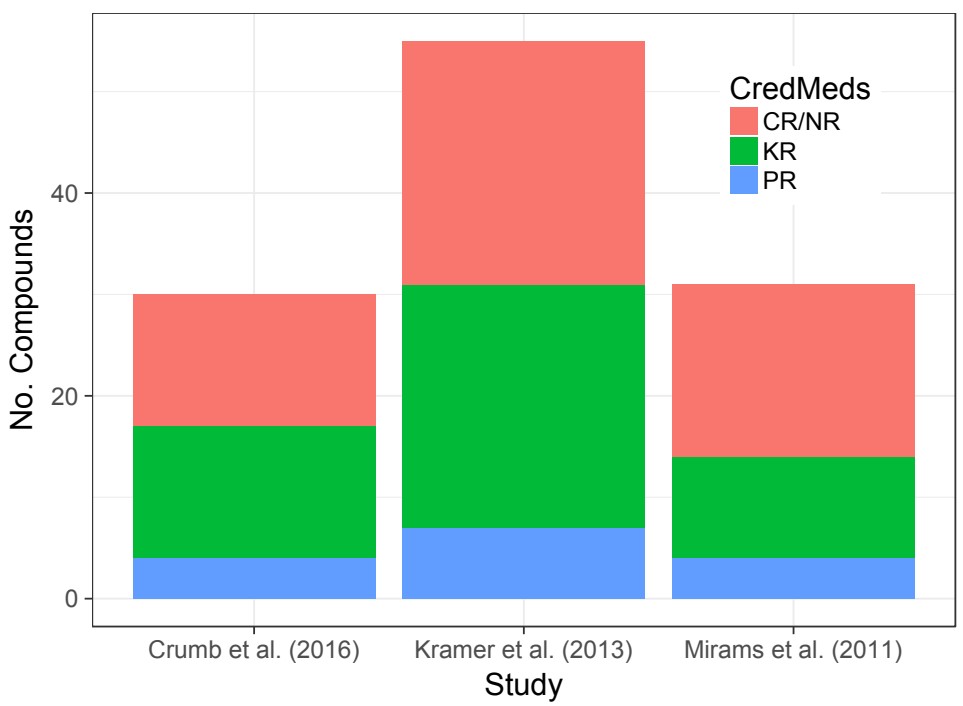

**Figure 1** Stacked bar-chart shows the proportion of compounds in each data-set that are KR, PR or CR/NR based on information within the CredibleMeds database.

**Table 2** ROC AUC values from the leave one out cross validation for assessing the joint QT/TdeP risk across all data-sets for all models considered.

| Data-set | Leave One Out Cross Validation ROC AUC | | | |
|---|---|---|---|---|
| | 3 ion-channels | | | hERG |
| | $B_{net}$ | ORD: ΔAPD90 | TT: ΔAPD90 | % Block IKr |
| *Mirams et al. (2011)* | 0.71 | 0.53 | 0.68 | 0.51 |
| *Kramer et al. (2013)* | 0.96 | 0.86 | 0.94 | 0.67 |
| *Crumb Jr et al. (2016)* | 0.71 | 0.65 | 0.65 | 0.61 |
| | 7 ion-channels | | | |
| *Crumb Jr et al. (2016)* | 0.82 | 0.67 | 0.60[*] | |

**Notes.**
[*]based on 6 ion-channels—INaL not modelled by TenTusscher et al. (TT); ΔAPD90: percentage change in APD90.

performance over just hERG block regardless of the classification scheme used. Note that again the performance of ORD is not as high as $B_{net}$ or TT. In addition the difference between $B_{net}$ and TT is negligible.

Within the latest data-set by Crumb et al. the performance of all models, when using only three ion-channels, drops to a level similar to that seen within the Mirams et al. data-set. The key difference between the results between those two data-sets is that ORD now shows similar performance to TT regardless of the classification scheme used. Furthermore, neither biophysical model performs overly better than using hERG block. $B_{net}$ however
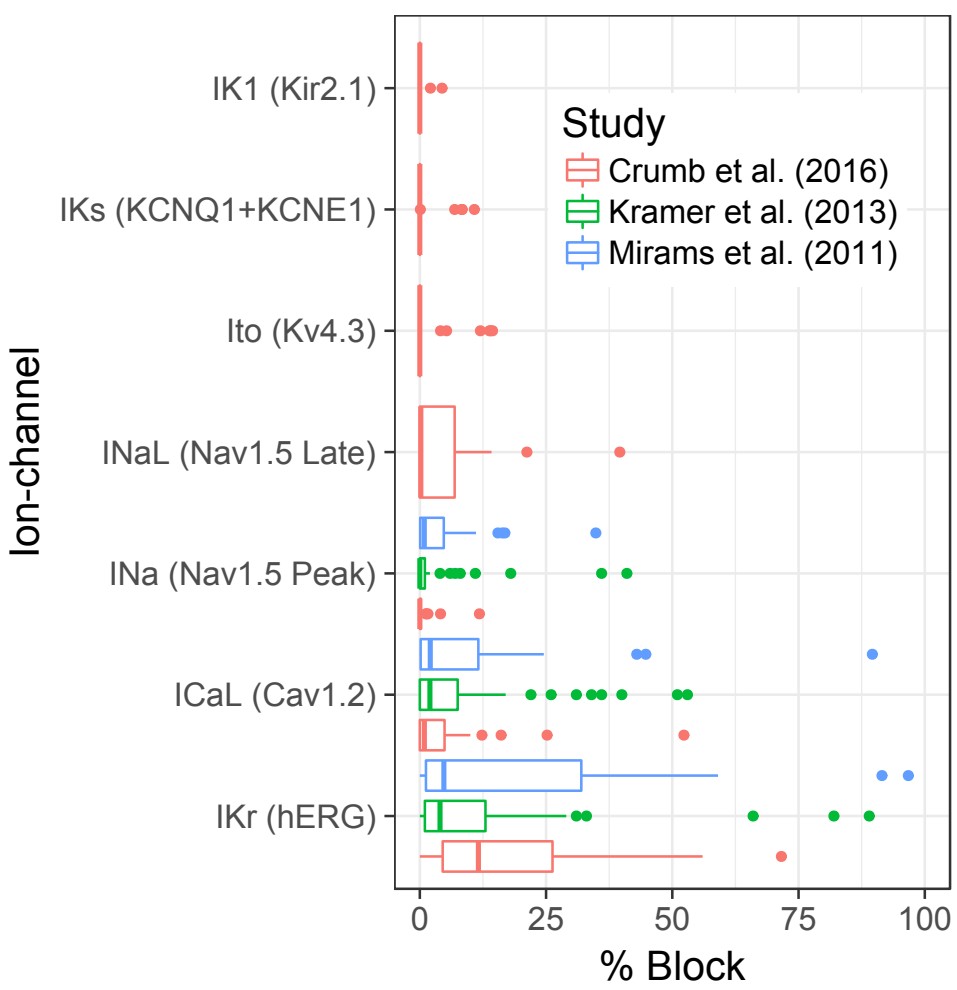

**Figure 2** Boxplots show the distribution of block for each ionic current across all 3 data-sets.

**Table 3** ROC AUC values from the leave one out cross validation ROC AUC for assessing TdeP risk only across all data-sets for all models considered.

| | Leave One Out Cross Validation ROC AUC | | | |
|---|---|---|---|---|
| **Data-set** | **3 ion-channels** | | | **hERG** |
| | $B_{net}$ | **ORD: ΔAPD90** | **TT: ΔAPD90** | **% Block IKr** |
| *Mirams et al. (2011)* | 0.78 | 0.66 | 0.75 | 0.62 |
| *Kramer et al. (2013)* | 0.86 | 0.80 | 0.84 | 0.68 |
| *Crumb Jr et al. (2016)* | 0.68 | 0.61 | 0.62 | 0.57 |
| | **7 ion-channels** | | | |
| *Crumb Jr et al. (2016)* | 0.77 | 0.63 | 0.59[*] | |

Notes.
   *based on six ion-channels—INaL not modelled by TenTusscher et al. (TT); ΔAPD90: percentage change in APD90.

appears to give reasonable performance again and appears to show an improvement over using hERG block for both classification schemes. Finally, when moving onto using all the ion-channel data from the Crumb et al. data set the difference in performance between the models is quite striking. $B_{net}$'s performance improves with the addition of more information whereas there is little improvement in either biophysical model.

In summary the results show that the performance of the models is data-set dependent. However, within each data-set the $B_{net}$ model performs just as well if not better than leading biophysical models.

## DISCUSSION

There appears to be a strong belief within the field of ion-channel cardiac drug toxicity that large scale single cell (*Mirams et al., 2011*) and even whole heart models (*Okada et al., 2015*) are required to answer a well-defined question: does measuring more than hERG improve prediction for TdeP, in humans? The evidence base, that suggests that large-scale biophysical models perform better than simpler models for this question, simply does not exist. Previous studies have shown that the performance of the large-scale cardiac models can be mirrored by simpler models (*Mistry, Davies & Di Veroli, 2015*; *Mistry, 2017*).

This study builds on those previous studies (*Mistry, Davies & Di Veroli, 2015*; *Mistry, 2017*) of comparing the performance of complex biophysical models versus simpler models in the following way. First a consistent definition of TdeP risk based on the CredibleMeds database was used across all data-sets (*Woosley, Heise & Romero, 2017*; *Woosley et al., 2017*). Second in addition to data-sets that considered drug activity against only 3 ion-channels (*Mirams et al., 2011*; *Kramer et al., 2013*) a third data-set (*Crumb Jr et al., 2016*) which measured drug affinity against seven ion-channels was also assessed.

The three models evaluated in this study were: (1) the *gold-standard* (*Zhou et al., 2015*) single cell model by *O'Hara et al. (2011)*; (2) the single cell model by *Ten Tusscher & Panfilov (2006)* which is used within the *cardiac safety simulator* (*Glinka & Polak, 2015*); (3) a linear mechanistic model evaluating the net difference in block between ion-channels involved in repolarising and depolarising the action potential, $B_{net}$ (*Mistry, 2017*). Each model was assessed via a leave-one-out cross validation using two different classification schemes based on the CredibleMeds database. The first scheme focussed on the joint QT prolongation and TdeP risk whereas the second scheme focussed on TdeP risk only. In addition to using outputs from the aforementioned models within the classification exercise the amount of block against hERG channel was used as a naïve benchmark.

Overall the analysis conducted showed that the performance of $B_{net}$ was superior to the more complex cardiac models regardless of the classification scheme used. $B_{net}$ was also the only model that consistently showed the benefit of measuring more than hERG. Finally $B_{net}$ was the only model whose performance improved when moving from using information against three ion-channels to seven. These results may appear surprising but are not uncommon in prediction problems in other fields (*Makridakis & Hibon, 2000*; *Green & Armstrong, 2015*). The key reason why complex models are not necessarily more predictive than simpler models is due to model error i.e., error in the structure of the model

itself (*Beven, 2005*). The concept of model error has only recently been assessed (*Beattie et al., 2017*) within the cardiac modelling field and so more needs to be done. Thus, the effect of model error on predictivity is largely unknown, although in other fields it tends to dominate prediction uncertainty (*Orrell et al., 2001*; *Refsgaard et al., 2006*).

This study is not without its caveats. The first is that the data-sets used may be too small to understand how large a discrepancy there truly is between the different models. However it is hoped that by continuing to assess new data-sets as they become available that the community will eventually have a comprehensive compound list. Second, the latest data-set by *Crumb Jr et al. (2016)* although measured the affinity of drugs against seven ion-channels the compounds only really showed activity against three. Thus, whether the results seen here will hold for a set of compounds with activity against a large number of ion-channels still remains unknown. Similar to the previous caveat this can only be assessed as more data is generated. The final caveat relates to the $B_{net}$ model itself. The model currently doesn't consider the kinetics of blocking which has been highlighted as an important factor (*Di Veroli et al., 2014*). However, these studies have been on a small numbers of compounds and so a true assessment of the importance of kinetics cannot be determined from those studies alone. If sufficient evidence regarding the importance of drug kinetics does eventually become available, adjustments to the $B_{net}$ model could be made.

## CONCLUSION

In summary, the study conducted here highlights the importance of benchmarking complex models against simpler ones. Furthermore, it highlights that simple mechanistic models can not only give similar performance to large-scale mechanistic models but can out perform them. Finally, it is hoped this study highlights that there is more than one solution to a problem and that although the question and quality of data dictates the modelling approach it should not dictate the size of the model.

### Funding
The author received no funding for this work.

### Competing Interests
The author declares there are no competing interests.

### Author Contributions
- Hitesh B. Mistry conceived and designed the experiments, analyzed the data, contributed reagents/materials/analysis tools, wrote the paper, prepared figures and/or tables, reviewed drafts of the paper.

### Data Availability
   The raw data is provided as Data S1.

## Supplemental Information

Supplemental information for this article can be found online at http://dx.doi.org/10.7717/peerj.4352#supplemental-information.

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
