# Peer review of "Complex versus simple models: ion-channel cardiac toxicity prediction"

_PeerJ, doi:10.7717/peerj.4352_

## Round 0.1 · original submission · Minor Revisions

Dear Hitesh,

thank you for submitting your manuscript to PeerJ. Your paper has been evaluated by two experts, who both appreciated the quality and importance of the manuscript. Please work through the very insightful suggestions that will help improve the impact of your manuscript.

Best wishes,

-Heiko

·

Basic reporting

Line 64, give a sentence explaining that torsades de pointes is. This should help explain why you use APD90 later.

Line 87, please give some information about Credible Meds. Why is their rating the one you chose? Are they considered the gold standard by the field? Are there other ratings? If so, what are the advantages and disadvantages of these other ratings and why were the other rating systems not chosen?

Line 100, change “the reduces the flow” to “that reduces the flow”

Line 105, change “known (KR)” to “known risk (KR)”

Line 112 – How was the EFTPC calculated? Was it the average concentration for the clinically approved dose? Was protein binding taken into account when comparing in vitro and clinically effective concentrations?

Line 117, What is the AP predict platform? Briefly describe it in 1-2 sentences.

Line 128, Why is your parameter called Bnet? Is it because it’s the “net difference” between the repolarization and depolarization block? If so, consider using the words “net difference” in the definition of Bnet so that the reason for the name is clear.

Line 131 – which ion channels are polarizers and depolarizers?

Line 132 – Where did the idea to use APD90 come from? Have other metrics been used in the past prediction? Would the results be expected to change with a different threshold, say APD80? Or are there other metrics one might consider?

Line 140 – An algorithm is described for using the complex model to assess risk, using the logistic regression presented by Cummins Lancaster and Sobie. Can the author explain who these people are? Did they develop either the gold standard model or the cardiac safety simulator? Is this the only method that has been used to predict cardiac toxicity risk or are there others? Is there consensus from the community that this metric is best?

Line 155, Line 207, Table 1 caption – add the words “ROC AUC” to make the phrase “leave-one-out cross validation ROC AUC” for clarity

Figure 2 – This figure is a bit hard to read. It is hard to compare across panels because there is a different coloring and order for Crumb as for the other two. For instance, iKr is red in two plots and yellow in the third. I would propose the author redraw this figure so that all the key features of the data described in the text stand out in the figure. I suggest
1) Use just a single panel instead of 3 panels.
2) Change the shape and color of the points based on the reference (Mirams, Kramer, Crumb)
3) Sort the ion channels by variability rather than alphabetical order, such that the channel with the greatest variability (iKr) appears first
4) I also suggest making this a forest plot where the ion channel labels are along the y-axis and the % block along the xaxis. In this way, both names for the receptors [e.g. hERG (iKr)] can be printed and easily read. This would be helpful for readers who may have some passing familiarity with hERG, but don’t know the “iKr” nomenclature.

Line 183 – delete the word “As” – currently this is a run-on sentence

Line 185 – I propose stronger language, deleting the words “has the potential to” because the key result is that the Bnet model does outperform the others.

Line 187-200. This text belongs in the introduction as this text all describes background results from two previous studies. The author could still briefly recap what this work adds over the previous studies in the Discussion section.

Line 187 – what are “those” previous studies? Is it Colatsky and Mistry from the previous paragraph? Please cite the two studies again, just for clarity.

Line 192 – The author is talking about the advantage of CredibleMeds to other classification schemes, but the author hasn’t described these other classification schemes. What are the other schemes called? Who has used them in the past? And do the results change depending on the classification scheme? Are there any other advantages to these schemes over CredibleMeds?

Line 207 – I don’t understand why the predictive accuracy of these three models cannot be prospectively tested. Couldn’t they still be tested for drugs that are predicted not to have a TdeP risk and then either do or do not show this risk in the clinic.

Line 212 – replace “if not” with “or”

Line 213-221 – I propose deleting all but the first sentence of this paragraph because the rest of the paragraph just restates in detail all the results that were just presented in the Results section. All that is needed here is a summary of the key message.

Line 237 – need comma after “available”

Line 238 – I don’t understand what it means to make the Bnet variables time dependent. How would one then derive a single metric from this?

Line 243 – Propose adding the following text (demarked with asterisks) In summary the study conducted here highlights the importance of benchmarking *complex models against simpler models*.

Experimental design

Line 192 If feasible, I’d ask the author to consider generating a new Figure with the results for alternative classification schemes (to CredibleMeds) to see if the results stay the same, or if a different model becomes the winner. If this is not possible, (because the other classification schemes are not available for all compounds on the other studies, or for other reasons) then please say why it’s not possible.

Validity of the findings

no comment

·

Basic reporting

This article reports what has been done well, it is clear and unambiguous. I was able to reproduce all of the results that I tried to using the supplementary data files.

In terms of how the biophysically-detailed models are presented (‘gold-standard’ and ‘cardiac safety simulator’) I strongly suggest using their names instead (‘O’Hara 2011’ and ‘Ten Tusscher 2006’).

In the first case, whilst O’Hara may be a leading model some of its strongest advocates are optimising it for studies of drug action, so do not consider it a gold standard. I know that this term has been taken from a previous study, and that the use of italics may imply some questioning of that name, but I don’t think these subtleties will come across and it would be preferable to call it by its original name. (Incidentally updated versions of O’Hara have been published recently by the FDA team, I would be happy to provide a CellML file for this which we are working on testing at the moment and will publish shortly, if the author would like to re-run with the ‘CiPA v1.0’ O’Hara model).

In the second case, whilst the commercial ‘Cardiac Safety Simulator’ uses Ten Tusscher 2006 it is not the Cardiac Safety Simulator model. The model in this study is simply the Ten Tusscher 2006 model, and was used for exactly this purpose in Mirams et al. Cardiovasc. Res. (2011). It would be better to simply call it Ten Tusscher (2006). I think the Cardiac Safety Simulator provides some additional endpoints in terms of a contraction model and electro-mechanical window etc. too.

In the introduction it is stated that “including more detail…the degree of structural uncertainty [increases]”. This sounds like common sense: at some point I agree there will be too much detail. But up to a point, the complexity may be warranted, parameterisable, and 'selectable' as a structure. Consider for instance a Closed-Open model for an ion channel. I can easily show this model is too simple for some channels, and a Close-Open-Inactive model is better. i.e. I am more certain in a more complex structure with more parameters. As we continue to add states, your statement is true, but there is a 'happy medium' we probably both agree the community should turn our attention to identifying for this application. I think this introduction risks over-simplifying the subtleties of model construction and selection.

Experimental design

Whilst the Crumb dataset does cover 7 ion channels, it does not feature significant activity at many of them, and is actually not dissimilar to considering only 2-3 ion channels. I wouldn’t strongly make a statement in any study based on that dataset that the information on the extra ion channels is interpreted better by a given model over another.

If a strong IK1 blocker, or Ito blocker was encountered I would still have more confidence in say a classifier based on Ten Tusscher 2006 APD predictions than a classical black box classifier. Where Bnet falls in this I am not entirely sure, but I would not be as confident as the author appears to be that it would perform as well. Perhaps some more limitations in terms of the ‘regimes’ that have been tested would be useful (Figure 2 goes some way to looking at this, but it is really an N-D dataset and is difficult to see how similar these datasets really are, or how much of the possible-block space they cover).

Validity of the findings

There is something of a problem with using Credible Meds for this study. PR is defined (on the credible meds website) as: "Possible Risk of TdP - These drugs can cause QT prolongation BUT currently lack evidence for a risk of TdP when taken as recommended". Note that this means compounds classified as PR could be QT prolongers that are NOT pro-TdeP, raising concerns about the dataset we are using for testing here. Such compounds do exist and are a major reason for the recent efforts to address TdeP risk. In taking Credible Meds as a ‘gold standard’ there is a danger of testing all these measures for QT prolongation rather than pro-arrhythmic risk. In terms of how this might change the findings, it is conceivable that the subtle transition to pro-arrhythmic behaviour may be captured better by more biophyscially-detailed models, or maybe vice-versa, but this article is tackling a slightly different question to the one that CiPA (www.cipaproject.org) is aiming to address.

Unfortunately I don't have a better suggestion at present, other than perhaps using the CiPA compound list where this distinction between QT and pro-arrhythmia was carefully considered and motivates the compound choice to some extent. But I acknowledge it is far more limited in number of compounds. It may be best just to expand the discussion of this testing set and acknowledge this explicitly as a limitation.

Additional comments

This is an interesting article and I believe the author is right to make the point that the use of biophysically detailed models in drug safety testing should be justified. The use of simple(r) models should set a benchmark that biophysically detailed models should have to beat to be widely adopted. I have written a blog on this myself, and agree with the main sentiment of the article: https://mirams.wordpress.com/2017/03/27/risk-regression-or-biophysics/

That said, in the article Discussion, I think it is worth stressing a few points in favour of biophysical models, that the author may wish to include in the discussion:
1. Bnet itself is really a simple biophysical model – by relying on Hill curves and a knowledge of electrophysiology it could be called biophysical.
2. In some sense Bnet is selected as it gives a good answer here, whilst the other models are operating a validation study. Whilst it is pleasingly simple and there is not much room for tweaking, the Bnet model structure itself was selected and is different to previous simple models the author has proposed. So there is an element of showing us the model that happened to work (which I will admit some simulation studies have also been doing recently), I don't think this is a massive issue as the Bnet model has no parameters, only structure, to tweak.
3. Many of the currents were discovered with the aid of the biophysical models (not that that means we should necessarily use them in this context, but the readers should be aware of it).
4. Chronologically, the big jump in predictive power - from hERG-only to multichannel block was actually performed because of the biophysically detailed models (including Ten Tusscher 2006 - labelled here as 'cardiac safety simulator'). The fact the models existed led us to try this, the multi-ion channel data had been available for some time before Mirams et al. 2011 proposed its use for improving TdP predictions because models suggested that combination of data would be useful!

3 and 4 aren’t necessarily a defence of keeping to using more detailed models for safety prediction if simpler models are shown to be as predictive, but they do strongly suggest that biophysically-based models may perform dimension reduction to a single 1D risk marker to deal better with ‘edge cases’ (combinations of ion channel block that have not been seen in training data for a simple classifier) than simpler models. Whether Bnet can also do this remains to be seen I would say.

I think the author should probably discuss and cite some of the whole-heart simulation studies that have been proposed for this purpose, as they are even further down the other end of the spectrum from the ones that he focusses attention on here.

I have made some more minor comments in direct annotations to the attached PDF.

---

## Round 0.2 · accepted · Accept

Dear Hitesh,

Thank you for your thorough revisions, and congratulations on the acceptance of your manuscript.

Best,
-Heiko

·

Basic reporting

The author addressed all my comments

Experimental design

No comment

Validity of the findings

No comment